# Developing and Validating Risk Scores for Predicting Major Cardiovascular Events Using Population Surveys Linked with Electronic Health Insurance Records

**DOI:** 10.3390/ijerph19031319

**Published:** 2022-01-25

**Authors:** Hsing-Yi Chang, Hsin-Ling Fang, Ching-Yu Huang, Chi-Yung Chiang, Shao-Yuan Chuang, Chih-Cheng Hsu, Hao-Min Cheng, Tzen-Wen Chen, Wei-Cheng Yao, Wen-Harn Pan

**Affiliations:** 1Institute of Population Health Sciences, National Health Research Institutes, Maoli 350401, Taiwan; sling0329@nhri.edu.tw (H.-L.F.); ccy@nhri.org.tw (C.-Y.C.); chuangsy@nhri.edu.tw (S.-Y.C.); cch@nhri.org.tw (C.-C.H.); pan@ibms.sinica.edu.tw (W.-H.P.); 2Institute of Public Health, National Yang-Ming University, Taipei 112304, Taiwan; hmcheng@vghtpe.gov.tw; 3Health Service Division, Industrial Technology Research Institute, Hsinchu 310401, Taiwan; CY.huang@itri.org.tw; 4Department of Cardiology, Taipei Veterans General Hospital, Taipei 112201, Taiwan; 5Department of Nephrology, Wei-Gong Memorial Hospital, Maoli 350401, Taiwan; twchen@tmu.edu.tw; 6Department of Pain Management, Min-Sheng General Hospital, Taoyuan 330056, Taiwan; m000924@e-ms.com.tw; 7Institute of Biomedical Sciences, Academia Sinica, Taipei 115201, Taiwan

**Keywords:** National Health Insurance database, risk prediction, population survey

## Abstract

A risk prediction model for major cardiovascular events was developed using population survey data linked to National Health Insurance (NHI) claim data and the death registry. Another set of population survey data were used to validate the model. The model was built using the Nutrition and Health Survey in Taiwan (NAHSIT) collected from 1993–1996 and linked with 10 years of events from NHI data. Major adverse cardiovascular events (MACEs) were identified based on hospital admission or death from coronary heart disease or stroke. The Taiwanese Survey on Hypertension, Hyperglycemia, and Hyperlipidemia (TwSHHH), conducted in 2002 was used for external validation. The NAHSIT data consisted of 1658 men and 1652 women aged 35–70 years. The incidence rates for MACE per 1000 person-years were 13.77 for men and 7.76 for women. These incidence rates for the TwSHHH were 7.27 for men and 3.58 for women. The model had reasonable discrimination (C-indexes: 0.76 for men; 0.75 for women), thus can be used to predict MACE risks in the general population. NHI data can be used to identify disease statuses if the definition and algorithm are clearly defined. Precise preventive health services in Taiwan can be based on this model.

## 1. Introduction

Precise preventive health services emphasize managing chronic diseases and developing individualized risk prediction to improve the quality and effectiveness of health care [1]. The ideal disease prevention should start with screening risk factors for predicting risk and then promote health throughout a patient’s entire life. Cardiovascular disease is a major cause of death and disability worldwide. The World Health Organization (WHO) recognizes the seriousness of cardiovascular disease (https://www.who.int/cardiovascular_diseases/guidelines/Pocket_GL_information/en/, accessed on 30 December 2021) and suggests that most cardiovascular diseases (CVDs) are preventable. A pocket guide can be used to identify people at high risk and provide guidance to prevent heart attacks or strokes. In the U.S., CVD risk is estimated using the Framingham Risk Score (FRS), which was developed in the U.S. when it was facing a high CVD prevalence [2,3]. The ability of the FRS to accurately predict CVD risk for different ethnicities is uncertain. Individual studies examining FRS performance have been conducted in Europe [4,5,6,7] and Asia [8,9,10,11,12]. Knowing the limitations of the original FRS, the American Heart Association (AHA) and the American College of Cardiology (ACC) released a new risk calculator [13]. The calculator, known as the ASCVD, was developed by pooling data from several cohorts, including cohorts from the Atherosclerosis Risk in Community [14], the Cardiovascular Health [15], the Coronary Artery Risk Development in Young Adults [16], and the Framingham, Original [17] and Offspring [18] studies. In November 2013, the AHA and ACC updated the clinical guidelines for managing lipids [19], and risk score was an important feature. The International Atherosclerosis Society provided a calibration factor for each country with different risks [20]. Following their instructions, we generated and submitted our own calibration factor [21]. Based on our experience in generating the calibration factor and observations from Asian countries [8], risk factors for coronary heart disease (CHD) may differ in Asian populations. Liu et al. recalibrated the Framingham prediction function using data from the Chinese Multi-Provincial Cohort study (CMCS) [11]. We applied their new coefficients to our national representative data from the 2002 Taiwan Survey of Hypertension, Hyperglycemia, and Hyperlipidemia (TwSHHH) and found that the prediction did not fit the Taiwanese population [22]. Thus, we developed our own model. 

Most of the scores were based on longitudinal follow-up, and the diseases were ascertained either from self-reports or doctors’ diagnoses. Periodic follow-up of patients nationwide would be very costly. As use of electronic medical records (EMRs) increases, one form of the EMR can be used to ascertain events. Since 1995, the National Health Insurance (NHI) program has provided universal coverage to more than 99% of the population in Taiwan. To get reimbursed, medical facilities (either clinics or hospitals) must upload the information for each patient’s clinical or hospital visits. Thus, the NHI database contains personal characteristics (sex, date of birth, and insurance information) and clinical information (date, expenditures, diagnosis, prescription details, and operations) [23]. We conducted this study to construct and validate our own MACE risk prediction model using national surveys linked to claim data and the death registry.

## 2. Materials and Methods

### 2.1. Data

Data for developing the model were obtained from a nationwide survey, the 1993–1996 Nutrition and Health Survey in Taiwan (NAHSIT), which asked questions on disease history, food intake, and health-related behaviors. Participants were asked to fast for 8 h before physical examination. Blood samples were centrifuged immediately. Serum samples were frozen on dry ice, then delivered to the Academia Sinica and frozen at −70 °C on the same day. Frozen serum samples were analyzed in a certified laboratory. Coefficients of variation from the blood samples were within acceptable ranges deriving from 5% split blood samples [22]. No written consent form was required during the survey; however, we obtained patients’ oral consent before the survey. The 1993–1996 NAHSIT was the first representative survey with biomarkers. It was also conducted during the time National Health Insurance (NHI) was implemented. It was the beginning of following-up individuals using nationwide insurance claim data electronically.

Validation data were obtained from another nationwide survey, the TwSHHH, conducted in 2002 and followed up in 2007 [24]. The health examination was similar to that of the NAHSIT; however, nutrition intake was not recorded. Participants were informed about the survey, and data were linked only from those who signed the consent form. 

The model included the following data: baseline values of log (age), systolic blood pressure (SBP), diastolic blood pressure (DBP), fasting glucose, total cholesterol, high-density lipoprotein (HDL-C), low-density lipoprotein (LDL-C), ratio of total cholesterol to HDL-C, triglycerides, uric acid, body mass index (BMI; kg/m^2^), waist circumference (cm), waist-to-hip ratio, and smoking status. The Institute Review Board of the National Health Research Institutes approved this study.

### 2.2. Events

MACEs are a group of disorders of the heart and blood vessels (https://www.who.int/news-room/fact-sheets/detail/cardiovascular-diseases-(cvds), accessed on 30 December 2021). Heart attacks and strokes are usually acute events and are mainly caused by a blockage that prevents blood from flowing to the heart or brain. MACEs, which include CHD and stroke, were extracted from the NHI databank or death registry. Only hospitalization or death due to CHD or stroke was considered. CHD was diagnosed as per ICD-9: 410-414 or ICD-10: I20-I25; stroke was diagnosed as per ICD-9:430-438 or ICD-10: I60-I69. Time-to-event was calculated from the date of the survey to the date of the event.

### 2.3. Statistical Methods

The model’s discrimination ability was evaluated using Harrell’s C [25], which evaluates the proportion of concordant pairs over all possible pairs. The formula is (Equation (1)),
(1)$$C=pr\left\{z\left(X_{i}\right)>z\left(X_{j}\right)|T_{i}<T_{j}\;\&\;D_{i}=1\right\}=\frac{\pi_{\mathrm{conc}}}{\pi_{\mathrm{comp}}}$$
where (Equation (2))
(2)$$\pi_{\mathrm{conc}}=pr\left\{z\left(X_{i}\right)>z\left(X_{j}\right)\;\&T_{i}<T_{j}\;\&\;D_{i}=1\right\}$$
represents the concordant pair, and (Equation (3))
(3)$$\pi_{\mathrm{comp}}=pr\left\{T_{i}<T_{j}\;\&\;D_{i}=1\right\}$$
represents all pairs. If a variable increased the C value by 0.002, the variable was kept in the model. The Akaike information criterion (AIC) was used to guarantee the goodness of the model fit. The model with the lowest AIC was selected. In other words, the model was selected based on a higher C value and lower AIC. The Hosmer–Lemeshow test was used to evaluate the calibration [26]. This test divided the predicted risk into ten groups then compared the observed risk to the predicted risk. The χ^2^ test was also used. The same method was applied to the external calibration. All analyses were conducted using R and SAS statistical software, version 9.4 (SAS Institute Inc., Cary, NC, USA).

## 3. Results

To build the model, we used data from 1658 men and 1652 women aged between 35 and 70 years. For each disease, we excluded those who reported having the disease at baseline. The MACE incidence rates were 13.77 per 1000 person-years for men and 7.76 per 1000 person-years for women. We constructed separate models for CHD and stroke, and then combined them as MACE. Table 1 compares the baseline variables between patients who developed MACE, CHD, or stroke and those who did not. Men who developed MACEs were significantly older, with higher blood pressure, higher waist-hip ratios, higher waist circumferences, and higher proportions of hypertension than their counterparts at baseline. Among women, almost all variables differed except the glucose level. The 10-year event probabilities S_0_(10) for MACEs were 0.82 for men and 0.90 for women. Thus, approximately 82% of the men and 90% of the women remained MACE-free during the 10-year period.

Table 2 presents the final models. The C values for CHD were 0.73 for men and 0.82 for women. The AIC values were the lowest in both models, at 760 for men and 474 for women. Agreement between the predicted and observed probabilities was classified into deciles and tested via a *χ*^2^ test. The *χ*^2^ were 22.91 for men and 21.68 for women. The C values for stroke were 0.80 for men and 0.79 for women and for MACEs were 0.76 for men and 0.75 for women.

Models were validated using TwSHHH data linked to the NHI data. The MACE incidence rates were 7.27 per 1000 person-years for men and 3.58 per 1000 person-years for women in the 1993–1996 NAHSIT sample. The 10-year MACE-free probabilities were 0.90 for men and 0.95 for women. Figure 1 shows the validation. The *χ^2^* statistics for comparing the predicted to the observed values were <20 for CHD and MACEs, but slightly higher for stroke (28.67 for men; 20.93 for women).

## 4. Discussion

In this study, we used data from national surveys linked to the NHI and death registry and extracted events over a 10-year period to develop a risk prediction model. Millions of records were processed. Manuel et al. used population health survey data to develop and validate a model for cardiovascular disease in Canada [27]. They combined the 2001, 2003, and 2005 Canadian Community Health Surveys to develop the model and used the 2007 survey to validate the model. The predictors were the self-reported risk behaviors, and the events were obtained from either the hospital admission for the disease or the death registry. We used a similar method to identify events, but chose surveys with biomarkers. Because the NHI data were for insurance claims, the data contained only the date of the claim and no information on disease conformation. We were advised by experts in the fields on hypertension, heart disease, and cerebrovascular disease to use the records from hospitalization or death to guarantee that real events were selected. 

The Framingham score and CMCS used categorical data to calculate the risk scores (Table 3). The Framingham score used blood pressure, total cholesterol, HDL-C, diabetes (yes/no), and smoking (yes/no) as predictors. The CMCS was developed using Chinese data [9]. Figure 2 shows the comparisons of the two models fitting the NAHSIT 1993–1996 data for CHD events. The C statistics were lower in men than in women, whereas the *χ^2^* statistic was much higher in women when using the Framingham score.

Chien et al. developed a point-based prediction model for CHD in Taiwan [28] using data collected in northern Taiwan. CHD was ascertained by physicians. The authors developed three models: a clinical model, a total cholesterol-based model, and an LDL-C-based model. The clinical model included age, sex, BMI, SBP, smoking status, and family history of CHD. The total cholesterol-based model was similar to the LDL-C-based model, except for the total cholesterol. However, the study lacked a nationally representative sample. Our sample was selected using a probability sampling scheme and covered the entire population of Taiwan. Therefore, our model can be used for risk prediction for the population in Taiwan. 

The first global estimate of the burden of 135 diseases listed cerebrovascular diseases as the second leading cause of death after ischemic heart disease [29]. The WHO reported that 15 million people suffer a stroke worldwide annually (http://www.emro.who.int/health-topics/stroke-cerebrovascular-accident/index.html, accessed on 30 December 2021). Approximately one-third of these remain disabled for long periods, resulting in heavy burdens on their family and community. Taiwan is no exception to this. The earliest risk prediction model for stroke was developed using the Framingham study in 1991 [30]. A risk prediction model was developed using Taiwanese community data in 2010 [31]. The incidence was ~6.8% in the 16-year follow-up. Two models have been developed based on these data [31]. One was a clinical model that included measures of blood pressure and disease history. The other was a biomedical model that included total cholesterol, white blood cell counts, and fasting glucose in addition to items in the clinical model. A model was developed for Chinese populations using the China Health and Nutrition Survey [32] using the incidence between 2009 and 2015. Separate models were developed for ischemic stroke and hemorrhagic stroke. Each population has different risk factors. Thus, we developed our model using Taiwan population data. We focused on severe stroke that resulted in hospitalization or death. Our model selected SBP, triglycerides, glucose, and uric acid for men and SBP, waist-hip ratio, smoking, hypertension, and diabetes for women. 

Because CHD and stroke are major cardiac events, we put them in one model. Our final model included age, sex, SBP, waist-hip ratio, HDL-C, and uric acid for men. The weight (coefficient) was heavy for the waist-hip ratio, implying that obesity may contribute largely to MACEs in men. Waist circumference, SBP, total cholesterol/HDL ratio, and smoking were used in the model for women. Lipid profiles (total cholesterol/HDL ratio) played a relatively important role in MACE development in women. The C statistics reached 0.76 for men and 0.75 for women. Using another national survey with lower incidence rates to validate the models, the C statistics were higher than those in the original population, reaching 0.78 for men and 0.79 for women. The overestimation on risk scores has been observed in many models for the same purpose. The WHO CVD Risk Chart Working Group suggested it might be models were developed using incidence at the population level and might include recurrent cases [33]. We used event-history model and the National Health Insurance data, which only count the event once. The overestimation was high in the highest 10th percentile. It was possibly caused by the linear function in the model. We have tried other functions, but they did not improve the model. In the end, the purpose was risk prevention. A higher estimation might alert individuals to modify their lifestyle in order to lower their risks. 

This study had some limitations. First, no behavioral variables other than obesity-related variables were selected; thus, these variables may have all been expressed in patients’ blood pressure or biomarkers. Mediation models may be one solution. Second, we did not stratify stroke into different subtypes. However, our purpose was primary prevention in the general population. We hoped this model would apply to government-funded health check-ups for people aged ≥40 years. Taiwanese residents get free health check-ups when they are ≥40 years old. Those aged between 40 and 64 years get free health check-ups every 3 years; those aged ≥65 years get free health check-ups annually. The health check-up could implement our mode into the report and inform people about their 10-year risk of MACE. There is a website developed for risk prediction as well as guidelines for prevention (https://cdrc.hpa.gov.tw/index.jsp, accessed on 1 January 2022).

## 5. Conclusions

In conclusion, linking national surveys to health insurance data enabled generating a MACE risk prediction model. Our model was validated using a dataset from another survey conducted a few years later and with fewer incidences. The models performed well, indicating that our model was valid regardless of time.

## Figures and Tables

**Figure 1 ijerph-19-01319-f001:**
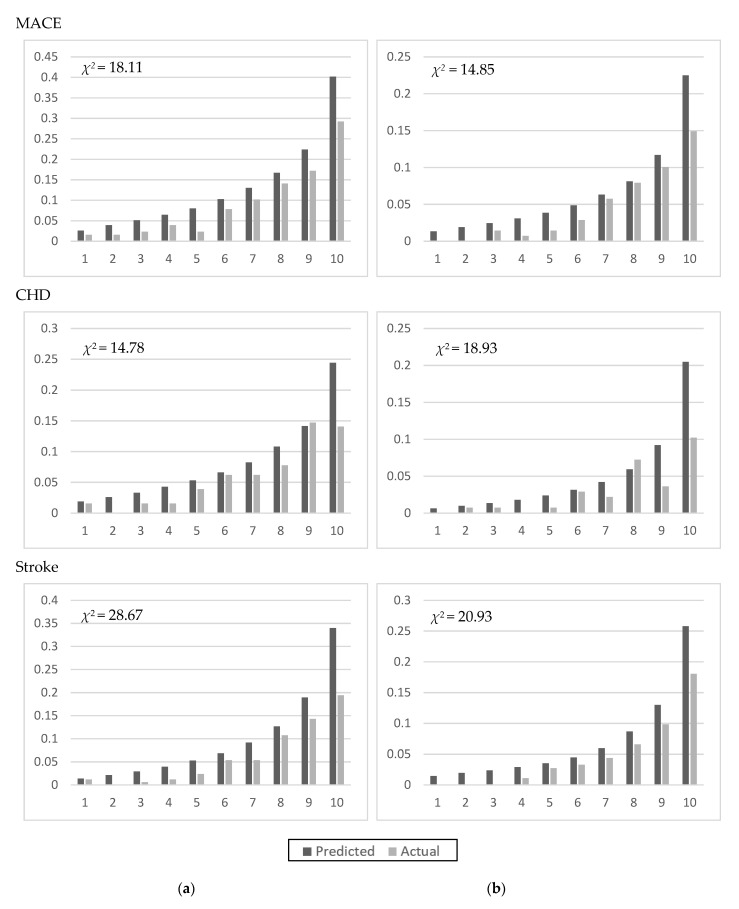
Validation of the developed models using TwSHHH data for men (**a**) and women (**b**).

**Figure 2 ijerph-19-01319-f002:**
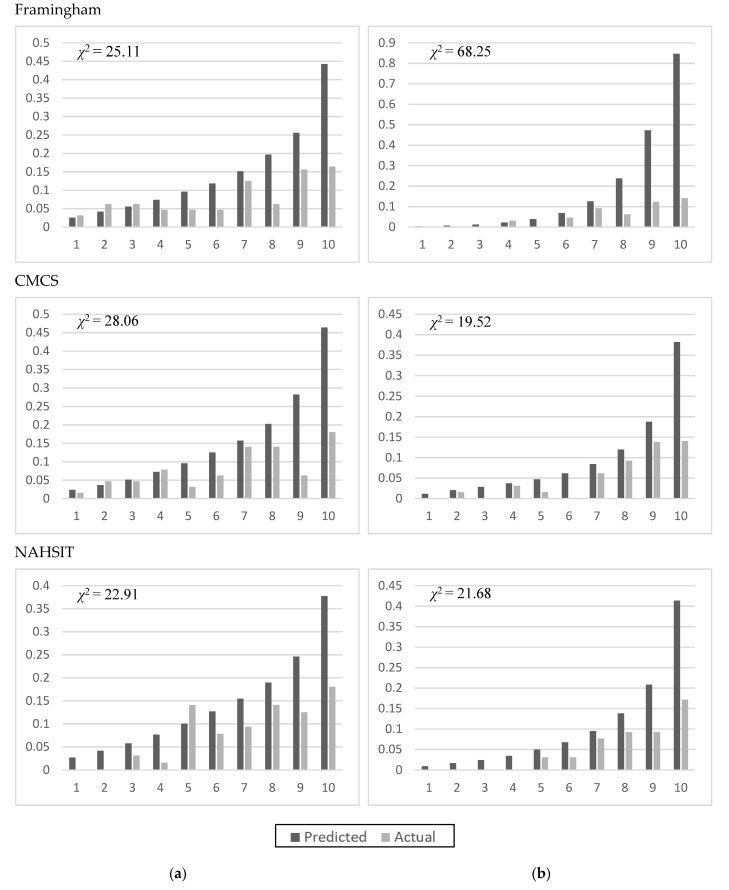
Applying different prediction models for CHD to the Taiwanese population (1993–1996 NAHSIT) for men (**a**) and women (**b**).

**Table 1 ijerph-19-01319-t001:** Comparisons of baseline values between patients who developed MACEs, CHD, or stroke over a 10-year period and those did not.

Men	MACE					CHD					Stroke				
	No (*N ^†^* = 1139)		Yes (*N* = 193)			No (*N* = 1230)		Yes (*N* = 102)			No (*N* = 1407)		Yes (*N* = 158)		
Variable	Mean	S.D.	Mean	S.D.	*p*-Value	Mean	S.D.	Mean	S.D.	*p*-Value	Mean	S.D.	Mean	S.D.	*p*-Value
Age (years)	51.06	9.93	57.82	8.58	*	51.59	9.99	57.42	8.95	*	52.41	10.11	58.65	7.88	*
SBP (mmHg)	126.59	17.02	139.50	23.22	*	127.79	18.12	136.48	22.30	*	128.90	18.42	142.34	21.76	*
DBP (mmHg)	82.15	12.08	85.89	14.34	*	82.62	12.52	83.61	12.19	0.44	82.86	12.20	88.04	15.05	*
Cholesterol (mg/dL)	196.69	37.12	194.86	39.07	0.63	196.32	37.19	197.61	40.04	0.79	197.42	37.80	193.85	37.72	0.37
Glucose (mg/dL)	96.42	18.15	97.60	21.09	0.59	96.44	18.13	98.29	23.37	0.56	99.87	26.33	107.68	37.77	0.06
Triglycerides (mg/dL)	124.71	79.88	133.84	80.79	0.28	125.06	79.84	137.02	81.91	0.27	128.56	80.56	139.91	87.00	0.21
HDL	54.05	17.91	53.32	28.81	0.79	54.13	19.99	51.71	18.94	0.36	52.97	17.82	53.23	29.69	0.93
LDL	117.17	36.54	116.31	35.67	0.82	116.99	36.39	117.64	36.64	0.89	118.28	36.77	113.91	36.75	0.27
Uric acid (mg/dL)	6.67	1.67	6.86	1.92	0.31	6.69	1.70	6.76	1.83	0.78	6.78	1.75	7.04	1.91	0.16
CHOL/HDL	3.98	1.35	4.22	1.68	0.16	3.99	1.35	4.34	1.96	0.17	4.10	1.48	4.17	1.43	0.66
BMI (kg/m^2^)	23.59	3.21	23.91	3.16	0.31	23.59	3.23	24.10	2.97	0.23	23.90	3.32	24.61	3.45	*
Waist-hip ratio	0.87	0.06	0.89	0.06	*	0.87	0.06	0.88	0.06	0.10	0.88	0.06	0.90	0.06	*
Waist (cm)	81.64	8.55	83.74	9.09	*	81.75	8.56	84.23	9.44	*	82.67	9.12	84.98	8.96	*
Smoker (%)	71		78		0.07	72		75		0.58	71		76	0.43	0.27
Diabetes (%)	3		6		0.18	4		5		0.55	7		17	0.38	*
Hypertension (%)	33		56		*	35		54		*	40		65	0.48	*
**Women**	**No (*N* =** **1116)**		**Yes (*N* = 106)**			**No (*N* = 1159)**		**Yes (*N* = 64)**			**No (*N* = 1473)**		**Yes (*N* = 121)**		
Age (years)	50.01	9.76	57.69	8.25	*	50.25	9.81	58.63	7.27	*	51.58	9.90	58.40	9.17	*
SBP (mmHg)	123.57	18.77	134.17	25.48	*	124.11	19.25	131.70	25.16	*	126.65	20.53	142.47	23.40	*
DBP (mmHg)	78.87	11.40	83.06	15.56	*	79.10	11.62	81.70	15.65	0.19	80.00	11.99	85.96	15.32	*
Cholesterol (mg/dL)	197.93	38.54	215.76	43.60	*	198.40	38.49	219.16	48.14	*	200.39	39.34	215.29	46.08	*
Glucose (mg/dL)	98.17	20.61	103.54	39.58	0.29	98.12	21.05	107.96	43.94	0.17	102.34	29.37	123.45	59.55	*
Triglycerides (mg/dL)	106.11	61.74	142.25	87.68	*	106.92	62.86	150.50	88.19	*	117.39	72.54	156.35	99.57	*
HDL	61.01	19.16	53.31	16.25	*	60.90	19.06	50.14	15.52	*	59.32	19.00	54.06	17.39	*
LDL	115.71	36.54	132.89	40.73	*	116.09	36.43	137.43	44.96	*	117.67	37.25	128.17	37.37	*
Uric acid (mg/dL)	5.31	1.45	5.67	1.57	*	5.32	1.47	5.75	1.40	0.06	5.54	1.62	5.98	1.88	*
Cholesterol/HDL	3.52	1.18	4.38	1.53	*	3.53	1.18	4.73	1.69	*	3.69	1.30	4.41	1.97	*
BMI (kg/m^2^)	24.25	3.69	25.78	4.07	*	24.29	3.73	25.94	3.75	*	24.71	3.76	25.83	4.33	*
Waist-hip ratio	0.80	0.07	0.84	0.07	*	0.80	0.07	0.85	0.07	*	0.80	0.07	0.85	0.07	*
Waist (cm)	76.00	8.88	81.43	9.85	*	76.13	8.97	82.85	8.99	*	77.39	9.32	81.94	9.42	*
Smoker (%)	6		9		0.34	6		6		0.86	6		17		*
Diabetes (%)	3		7		0.16	3		5		0.63	8		27		*
Hypertension (%)	27		44		*	28		39		0.06	36		64		*

^†^ All Ns excluded patients who reported having this disease at baseline. * *p* ≤ 0.05.

**Table 2 ijerph-19-01319-t002:** Variables used in the models for each disease by sex.

	Disease		
Men	MACE	CHD	Stroke
Incidence	13.77/1000	7.14/1000	9.53/1000
**Variable**	**Coefficient**	**Coefficient**	**Coefficient**
Age (years)	7.2782	8.3007	8.9606
SBP (mmHg)	0.0257	-	0.0231
Glucose (mg/dL)		-	0.0050
Triglycerides (mg/dL)		-	0.0013
HDL	−0.0039	-	
LDL		−0.0081	
Uric acid (mg/dL)	0.0214	-	0.0603
Cholesterol/HDL		-	
Waist-hip ratio	3.2778	-	
Waist (cm)		0.0163	
Smoke (yes)		-	
Diabetes (yes)		-	
Hypertension (yes)		0.6715	
C statistic	0.76	0.73	0.80
**Women**			
Incidence	7.76/1000	4.63/1000	6.98/1000
Age (years)	6.8833	9.3891	4.3538
SBP (mmHg)	0.0128	0.0015	0.0138
Glucose (mg/dL)			
Triglycerides (mg/dL)		0.0001	
HDL			
LDL			
Uric acid (mg/dL)			
CHOL/HDL	0.3054	0.3581	
Waist-hip ratio			3.9712
Waist (cm)	0.0257	0.0425	
Smoke (yes)	0.1923		0.7821
Diabetes (yes)			0.5348
Hypertension (yes)			0.5572
C statistic	0.75	0.82	0.79

**Table 3 ijerph-19-01319-t003:** Framingham and CMCS score coefficients.

	Men		Women	
Risk Factors	CMCS	Framingham	CMCS	Framingham
Age	0.07	0.05	0.07	0.17
Blood pressure				
Optimal	−0.51	0.09	−0.50	−0.74
Normal	Reference	Reference	Reference	Reference
High normal	0.21	0.42	−0.87	−0.37
Stage 1 hypertension	0.33	0.66	0.34	0.22
Stage 2–4 hypertension	0.77	0.90	0.47	0.61
Cholesterol, mg/dL				
<160	−0.51	−0.38	0.18	0.21
160–199	Reference	Reference	Reference	Reference
200–239	0.07	0.57	0.13	0.44
240–279	0.32	0.74	0.14	0.56
≥280	0.52	0.83	1.67	0.89
HDL, mg/dL				
<35	−0.25	0.61	0.62	0.73
35–44	0.01	0.37	0.30	0.60
45–49	Reference	Reference	0.08	0.60
50–59	−0.07	0.00	Reference	Reference
≥60	−0.40	−0.46	−0.78	−0.54
Diabetes	0.09	0.53	0.18	0.87
Smoking	0.62	0.73	−0.95	0.98

## Data Availability

The survey data and National Health Insurance data used in this study are available at the Ministry of Health and Welfare (https://dep.mohw.gov.tw/DOS/lp-2503-113-xCat-DOS_dc002.html, accessed on 30 December 2021).

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
