# Peer review of "Developing and Validating Risk Scores for Predicting Major Cardiovascular Events Using Population Surveys Linked with Electronic Health Insurance Records"

_ijerph, 2022, doi:10.3390/ijerph19031319_

Round 1

Reviewer 1 Report

Dear Authors,

The research paper entitled "Developing and validating risk scores for predicting major cardiovascular events using population surveys linked with electronic health insurance records" is well-written and thoroughly discussed.

Please explain the reason for designing the model using the Nutrition and Health Survey in Taiwan (NAHSIT) collected from 1993–1996.

Please cross check the statement "The statistics for comparing the predicted to the observed values were <20 for CHD and MACEs but slightly higher for stroke (28.67 for men; 20.93 for women)" (line 148 and 149). As per Figure 1, the predicted to observed values are <20 for MACEs and Stroke but slightly higher for CHD (28.67 for men; 20.93 for women). 

Please also discuss the probable reasons for slightly higher predicted to observed values CHD.

Author Response

The research paper entitled "Developing and validating risk scores for predicting major cardiovascular events using population surveys linked with electronic health insurance records" is well-written and thoroughly discussed.

Reply: We appreciated all the comments from reviewers. They help us improve the manuscript greatly.

Please explain the reason for designing the model using the Nutrition and Health Survey in Taiwan (NAHSIT) collected from 1993–1996.

Reply: The 1993-1996 NAHSIT was the first representative survey with biomarkers. It was also conducted during the time National Health Insurance (NHI) was implemented. It was the beginning to follow-up individuals using nationwide insurance claim data electronically. We added the statement in the text (lines 89-90).

Please cross check the statement "The statistics for comparing the predicted to the observed values were <20 for CHD and MACEs but slightly higher for stroke (28.67 for men; 20.93 for women)" (line 148 and 149). As per Figure 1, the predicted to observed values are <20 for MACEs and Stroke but slightly higher for CHD (28.67 for men; 20.93 for women). 

Reply: Sorry for the confusion. We double checked our original tables and figures and found mistakes in pasting them to the manuscript. We rearranged Figure 1 and revised the statement above the figure.

Please also discuss the probable reasons for slightly higher predicted to observed values CHD.

Reply: The overestimation on risk scores has been observed in many models for the same purpose. The WHO CVD Risk Chart Working Group suggested it might be models were developed using incidence at the population level and might include recurrent cases. We used event-history model and the national health insurance data which only count the event once. The overestimation was high in the highest 10th percentile. It was possibly caused by the linear function in the model. We have tried other functions. But they did not improve the model. In the end, the purpose was risk prevention. A higher estimation might alert individuals to modify their lifestyle in order to lower their risks. This was added to lines 227-234.

Reviewer 2 Report

The topic is important and this is a good paper with potential clinical applications using the developed model which they had it validated with different dataset which adds additional evidence to the developed model.

Paper is informative and well written. The study is well performed and data supports conclusion.

The text is properly referenced and clearly written.

Author Response

Thanks for the comment.